# Occlusive and Proliferative Properties of Different Collagen Membranes—An In Vitro Study

**DOI:** 10.3390/ma16041657

**Published:** 2023-02-16

**Authors:** Vishal Sehgal, Nisarat Ruangsawasdi, Sirichai Kiattavorncharoen, Sompop Bencharit, Prakan Thanasrisuebwong

**Affiliations:** 1Master of Science Program in Implant Dentistry, Faculty of Dentistry, Mahidol University, Bangkok 10400, Thailand; 2Department of Pharmacology, Faculty of Dentistry, Mahidol University, Bangkok 10400, Thailand; 3Department of Oral and Maxillofacial Surgery, Faculty of Dentistry, Mahidol University, Bangkok 10400, Thailand; 4Office of Oral Health Innovation, Department of Oral Rehabilitation, The James B. Edwards College of Dental Medicine, Medical University of South Carolina, Charleston, SC 29425, USA; 5Dental Implant Center, Faculty of Dentistry, Mahidol University, Bangkok 10400, Thailand

**Keywords:** collagen membrane, human gingival fibroblasts, barrier function, fibroblast proliferation, guided bone regeneration (GBR), implant dentistry

## Abstract

Different collagen barrier membranes come in various sources and crosslinking that may affect barrier function and tissue integration. This study investigated barrier function and tissue integration of the three different collagen membranes (Jason^®^: porcine pericardium, GENOSS: bovine tendon, and BioMend^®^ Extend: cross-linked bovine tendon) with human gingival fibroblasts. The barrier function and tissue integration properties were determined under confocal microscopy. Morphological characteristics were observed using scanning electron microscopy. Our results showed that all collagen membranes allowed a small number of cells to migrate, and the difference in barrier function ability was not significant. The cross-linked characteristics did not improve barrier ability. The native collagen membrane surfaces allowed evenly scattered proliferation of HGF, while the cross-linked collagen membrane induced patchy proliferation. Statistically significant differences in cell proliferation were found between Jason and BioMend Extend membranes (*p* = 0.04). Scanning electron microscope showed a compact membrane surface at the top, while the bottom surfaces displayed interwoven collagen fibers, which were denser in the crosslinked collagen membranes. Within the limitations of this study, collagen membranes of different origins and physical properties can adequately prevent the invasion of unwanted cells. Native collagen membranes may provide a better surface for gingival cell attachment and proliferation.

## 1. Introduction

Dental implants are widely used as one of the treatment options to replace missing teeth [1]. There are many factors that account for the success of dental implants. The quality and quantity of soft and hard tissues available have a major impact on the longevity of dental implants. Esthetics of peri-implant soft tissues frequently demand adequate bone support underneath [2]. Shortly after extraction, up to 50% of the bone width is declined during the first six months [3]. Therefore, soft and hard tissue augmentations are required at most sites prior to surgical implant placement.

Guided bone regeneration (GBR) is a common practice for sites with inadequate bone due to its predictable alveolar bone regeneration technique [4]. Barrier membranes are an important key for GBR to prevent an infiltration of epithelial cells and fibroblasts into the bony defect, where the bone is being regenerated at a slower rate than the lining soft tissues. Hence, barrier membranes must be properly selected to allow adequate time for the migration, proliferation, and differentiation of osteogenic cells during bone regeneration. The ideal properties of barrier membranes used in GBR include cell occlusivity, tissue integration, space maintaining ability, biocompatibility, and easy clinical manipulation. Thus far, there is no single membrane to achieve all these criteria [5,6]. Expanded-polytetrafuoroethylene (e-PTFE) membranes are non-resorbable membranes that are considered the gold standard in regenerative dentistry and have been extensively studied in GBR [7]. However, they require surgical removal and are associated with complications due to membrane exposures. Bioresorbable membranes were developed to overcome these limitations and avoid the need for surgical removal which is less invasive for the patient [7,8,9,10].

A common bioresorbable barrier membrane used in GBR is collagen membrane which is made of natural polymers. The most common natural polymers used for collagen membrane fabrication are collagen type I and type III [8]. Natural collagen is extracted by decellularization and removal of antigenic components from primary sources which include but are not limited to porcine skin, porcine pericardium, and bovine tendon [11,12]. Collagen-based products can be made of either purified collagen after extraction and scaffold fabrication or from decellularization and optional crosslinking where the collagen is not extracted and also keeps the natural 3D microstructure. Different processing of collagen in theory may affect cellular and tissue healing responses. The degradation rate of these bioresorbable membranes can vary based on the material composition, local pH at the surgical site, and the action of degrading enzymes such as endogenous collagenases [7,8,13]. A modification by crosslinking improves the mechanical properties of collagen membranes as it can stabilize the collagen matrix and increases resistance to resorption. Different methods can be used to form a crosslinked collagen membrane such as ultraviolet (UV) irradiation and chemical processing [11,12,13]. However, the crosslinked collagen membranes and their degradation products can have toxic effects to the surrounding environment which can lead to delayed angiogenesis, insufficient bone regeneration, and interfere with wound healing [14,15,16,17]. Ideally, the resorption rate of collagen membranes should match the rate of new bone formation, and the barrier function of the membranes must be maintained during this period [5,8,18]. The rough inner surface of collagen membranes should be porous and serve as a framework for the attachment of bone regenerating cells and angiogenesis [17,19,20,21,22,23].

Soft tissue integration on membrane surfaces by the attachment and proliferation of fibroblasts is necessary to limit membrane exposures and subsequent graft failure [24]. The process of soft tissue integration must begin immediately postoperatively as peak postoperative swelling and edema occurs early during the healing and can lead to wound dehiscence [25]. In theory, the outer smooth surface of the barrier membrane prevents soft tissue growth into the defect while serving as a scaffold for the attachment of fibroblasts for tissue integration [17,19,20,21,22,23].

Different commercially available collagen membranes report various degradation times and ability of maintaining barrier function and rapid soft tissue integration; however, to the authors’ knowledge, there are no publications directly comparing various collagen membranes of different origins and fabrication processes for the barrier function and soft tissue integration using an in vitro controllable condition.

The aim of this study was to evaluate the barrier function along with the attachment and proliferation of human gingival fibroblasts (HGF) on two native non-crosslinked and one crosslinked collagen membranes used in GBR. The non-crosslinked collagen membranes used in this study were Jason^®^ Membrane made of porcine pericardium and GENOSS Collagen Membrane made of bovine tendon, while the crosslinked collagen membrane used was BioMend^®^ Extend made of bovine tendon and chemically crosslinked using glutaraldehyde. The migration and proliferation of human gingival fibroblasts (HGF) was observed under confocal microscopy. The microstructures of all the membranes were investigated under scanning electron microscopy.

## 2. Materials and Methods

The human gingival fibroblasts, HGF-1, ATC.CRL 2014TM cell line, used in this study were provided by American Type Culture Collection (ATCC).

### 2.1. Collagen Membrane Preparation

The three commercially available collagen membranes used in this study were Jason^®^ Membrane (Botiss Biomaterials GmbH, Zossen, Germany), Collagen Membrane (GENOSS, Suwon-si, South Korea), and BioMend^®^ Extend (Zimmer, Warsaw, IN, USA). Each membrane was cut into squares of size 15 × 15 mm^2^. Three collagen membrane squares of all brands were used for the barrier function test, and another three membrane squares were used for the proliferation test. Another two membrane squares of each brand were observed for morphology under scanning electron microscope (SEM). The collagen membranes of each brand were distributed equally and randomly in each test. The cells used in this study were Human Gingival Fibroblasts (Human Gingival Fibroblast HGF-1, ATC.CRL 2014TM American Type Culture Collection, USA) and were stained with CytoPainter Cell Proliferation Staining Reagent—green fluorescence (Abcam, Cambridge, United Kingdom) as per the protocol described in the manufacturer’s instructions. Briefly, 1X of the dye working solution was added to the prepared HGF and incubated for 30 min at 37 °C. The dye working solution was subsequently removed. Prior to viewing the membranes under a confocal microscope (STELLARIS 5, Wetzlar, Germany), the nuclei of the HGF were further stained with 4′, 6-diamino-2-phenylindole (DAPI, Thermo Fisher Scientific, Waltham, MA, USA) and the cells were at an excitation/emission wavelength of 511/525 nm, respectively.

### 2.2. Barrier Function Test

Collagen membrane squares were fixed on to Cell Crowns™ 48 inserts to immobilize them. The membranes were checked for leakage prior to placing in the well plates using PBS loading. The Cell Crown™ was inserted in the 24 well plates in a way that the membranes did not touch the bottom of the well plates. A total of 1 mL of DMEM with 10% FBS and 1% penicillin/streptomycin was added to the well plates and 100 μL of the fibroblast suspension at a density of 1.0 × 10^4^ cells/mL in DMEM with 10% FBS and 1% penicillin/streptomycin was added directly on the upper smooth part of the collagen membrane squares in the Cell Crown™ inserts.

Cell culture insert with attached polyester (PET) membrane with a pore size of 8 μm (Corning, New York, NY, USA) was used as a control for the barrier function test. For the positive control, 1 mL of DMEM with 10% FBS and 1% penicillin/streptomycin was added to the well plates and 100 μL of the fibroblast suspension at a density of 1.0 × 10^4^ cells/mL in DMEM with 10% FBS and 1% penicillin/streptomycin was added directly on the upper part of the PET membrane. On the other hand, for the negative control, 1 mL of DMEM with 1% penicillin/streptomycin without FBS was added to the well plates and 100 μL of the fibroblast suspension at a density of 1.0 × 10^4^ cells/mL in DMEM with 10% FBS and 1% penicillin/streptomycin was added directly on the upper part of the PET membrane.

The Cell Crown™ and cell culture inserts in the well plates were incubated at 37 °C and 5% CO_2_ atmosphere for 3 days before viewing under confocal microscopy to detect and analyze any penetration of fibroblasts through the membranes. The fluorescence intensities of HGF penetrated to the opposite side of the membranes were recorded on each membrane square, and the mean fluorescent intensity (MFI) of each collagen membrane brand and PET membrane was calculated. Intergroup comparison of the barrier function test of the three collagen membrane brands and control PET membranes was performed using one-way analysis of variance (one-way ANOVA) at a confidence interval of 95%, and post hoc analysis was completed using Tukey’s multiple comparison test.

### 2.3. Fibroblast Proliferation Test

For the fibroblast proliferation test, the collagen membrane squares were fixed on to Cell Crowns™ 48 (Sigma-Aldrich, MO, USA) inserts to immobilize them. The Cell Crown™ was inserted in the 24 well plates (Corning, New York, NY, USA) and 100 μL of HGF suspension at a density of 5.0 × 10^3^ cells in Dulbecco’s Modified Eagle’s medium (DMEM, ATC.30-2002, Gibco, Paisley, UK) with 10% Fetal Bovine Serum (FBS) and 1% penicillin/streptomycin (Thermo Fisher Scientific, Massachusetts, USA) was loaded on the upper smooth part of the collagen membrane squares. Cells grown on the surface of polyethylene terephthalate membranes with a pore size of 0.4 μm (Sigma-Aldrich, Missouri, USA) in DMEM with 10% FBS and 1% penicillin/streptomycin were used as a positive control, while cells grown on the surface of polyethylene terephthalate membrane in DMEM with 1% penicillin/streptomycin without 10% FBS were used as a negative control. All cell culture plates were kept in an incubator at 37 °C and 5% CO_2_ for 3 days. On the third day, the surface of collagen membranes and polyethylene terephthalate membranes were viewed under confocal microscopy to evaluate fibroblast proliferation. Fluorescence intensity of HGF proliferation on each membrane square was recorded at 3 fixed points, and the MFI of each collagen membrane brand and polyethylene terephthalate membrane was calculated. Intergroup comparison of fibroblast proliferation of the three collagen membrane brands and polyethylene terephthalate membranes was performed using one-way ANOVA at a confidence interval of 95%, and post hoc analysis was completed using Tukey’s multiple comparison test.

### 2.4. Membrane Morphological Analysis

The surface porosity and collagen fibril characteristics of all collagen membrane brands were observed under scanning electron microscope (JSM-6610 series, JEOL, Tokyo, Japan). Briefly, a layer of palladium was coated on the collagen membrane samples using a sputter coater (Quorum Technologies, East Sussex, United Kingdom) at a thickness of about 10 nm. SEM photography was performed at 5 to 10 kV using 100X and 500X magnifications at the smooth outer (tissue) surface and the rough inner (bone) surface of all collagen membrane squares.

### 2.5. Statistical Analysis

Intergroup comparison of barrier function test and fibroblast proliferation test of the three collagen membrane brands and control membranes were performed using one-way analysis of variance (one-way ANOVA) at a confidence interval of 95%, and post hoc analysis was completed using Tukey’s multiple comparison test.

## 3. Results

### 3.1. Barrier Function Test

The tests were performed after three days of loading HGF directly on the three collagen membranes along with the control, PET membranes (Figure 1). Very few HGF penetrated the collagen membranes, and there was no statistically significant difference in the barrier function of these collagen membranes. In contrast, the positive control PET membranes revealed a greater number of penetrated HGF and were statistically significantly different compared to the negative control PET membrane and all three collagen membranes (*p* = 0.02, *p* < 0.01, *p* < 0.01, and *p* < 0.01 against negative control PET, Jason, GENOSS and BioMend Extend membranes, respectively).

### 3.2. Fibroblast Proliferation Test

After 3 days of loading fibroblasts on the three collagen membranes and control polyethylene terephthalate membranes, Jason and GENOSS collagen membranes exhibited uniformly scattered and densely proliferated fibroblasts across the entire membrane surfaces, similar to the positive control polyethylene terephthalate membranes. On the other hand, BioMend Extend collagen membrane displayed unevenly scattered and spaced proliferation of the HGF (Figure 2). There was a statistically significant difference in the ability of HGF proliferation between Jason and BioMend Extend membranes (*p* = 0.04). On the other hand, no statistically significant differences were found between Jason and GENOSS membranes, and GENOSS and BioMend Extend membranes. The ability to allow fibroblast proliferation of the positive control polyethylene terephthalate membrane surfaces were statistically significantly greater than all the negative control polyethylene terephthalate membranes and collagen membranes and displayed highly dense and uniform scattered spindle shaped fibroblast proliferation (*p* < 0.01, *p* = 0.03, *p* = 0.01, and *p* < 0.01 against negative control polyethylene terephthalate, Jason, GENOSS, and BioMend Extend membranes).

### 3.3. Morphological Analysis

The morphological analysis of the collagen membranes under SEM revealed compact nonporous upper surfaces, and even at higher magnifications, minimal surface porosities were seen (Figure 3a,b,e,f,i,j). The lower surfaces revealed dense, yet fine, interwoven collagen fibers in Jason membrane (Figure 3c,d), while the collagen fibers in the GENOSS collagen membrane appeared thicker, with patchy distribution and were loosely intertwined (Figure 3g,h). The collagen fibers on the lower surface of the crosslinked BioMend Extend membranes displayed thick and dense interwoven networks with minimal gaps between the fibers (Figure 3k,l).

## 4. Discussion

Various characteristics of collagen membranes have been available for GBR technique. However, the effect of different origins of fabrication and crosslinking of collagen fibrils of different collagen membranes have never been observed and compared in an in vitro setting for efficiency in GBR applications. This study therefore investigated the barrier function, tissue integration, and morphology of three commercially available collagen membranes including Jason membrane (native, porcine pericardium), GENOSS membrane (native, bovine tendon), and BioMend Extend membrane (crosslinked, bovine tendon). Our results demonstrated that the barrier function properties were similar between the different types of collagen membranes; however, native collagen membranes displayed better HGF proliferation properties. On the other hand, the SEM demonstrated a thick and dense interwoven fibrin network in the crosslinked collagen membrane compared with the non-crosslinked collagen membranes. The non-crosslinked membrane from porcine origin showed fine collagen fibrils which were densely intertwined, while the collagen fibrils in the non-crosslinked membrane of bovine origin showed thick fibers with patchy distribution and loose fibril networks.

Membranes used in GBR are required to maintain space for alveolar bone regeneration. The secluded space created by the membrane should provide adequate barrier function by preventing an early migration of the gingival cells before bone regeneration, which occurs at a slower rate [5]. Collagen membranes are naturally derived resorbable membranes which provide advantages not only to eliminate surgical removal, but also to improve biocompatibility superior to synthetic materials [5,26]. Different characteristics of collagen that are used to fabricate membranes have shown different efficiencies to provide barrier function depending on the degradation rate of the collagen [8]. All collagen membranes tested in this study showed fewer fibroblasts moving through the membranes compared to the control, which was the PET membrane in a setting without chemotactic agent promoting cell migration. Although the collagen membranes were observed for only 3 days after loading the HGF, they showed efficiency to maintain the barrier function despite passing through many manufacturing processes. Note that in regard to the barrier membrane testing, after the seeding of fibroblasts, the time point used in this study was set as a period of 3 days due to two major reasons. First, a three-day postoperative period is critical as the wound inflammation peaks around this time, which makes the wound susceptible to dehiscence. Second, after multiple pilot experiments, the positive control PET membranes allowed for cell migration and penetration on the third day, and hence, the third day was chosen since cell migration could be seen at this time point in the transwell assay. A transversal imaging was not needed in this study due the thinness of the membranes, which did not allow for the collection of the fibroblasts in the middle layers. Third, confocal microscopy used to view the membranes layer by layer demonstrated that the fibroblasts were only detected on either surfaces of the membranes, and not in between the outer layers. Crosslinking of collagen has been developed to extend the degradation time for as long as possible, but it has never been observed on the barrier effects [19,27,28]. The BioMend^®^ Extend membrane used in this study is a chemically crosslinked collagen formed by glutaraldehyde using fiber reconstitution technique, and it has claimed to provide barrier function effects up to 18 weeks. Although our results showed that the non-crosslinked collagen membranes had a similar ability to prevent fibroblast infiltration compared to the crosslinked membranes. However, a previous in vivo study showed the biodegradation of BioMend Extend is slower which means that the barrier function property may last longer than the non-crosslinked collagen membranes, which may be clinically significant [19]. Our results could indicate that for short periods of time, the crosslinking of collagen membranes did not improve the barrier function. Likewise, the source of the collagen membrane did not demonstrate any difference in barrier function. Our results are limited by the fact that the 3-day experimental period of observing the barrier function of the collagen membranes may not be representative of all clinical applications [28,29,30,31]. However, when excluding the environmental factors, all collagen membranes can primarily prevent the gingival cell migration, and future studies observing the duration of barrier function should be considered.

Tissue integration is a major property that prevents wound dehiscence, which is an unwanted outcome that delays tissue healing or impedes bone regeneration in GBR [29,30]. Membranes used for GBR should be able to allow gingival and fibroblast attachment and proliferation to prevent wound dehiscence. The collagen membrane surfaces mimic epithelial basement membranes of extracellular matrix (ECM) as well as function as a scaffold that may play a role in facilitating fibroblastic attachment [32,33]. Better cell compatibility leads to better tissue integration. Native collagen membranes can be modified to enhance the durability and strength, and various methods have been proposed to increase covalent bond between collagen fibrils. Glutaraldehyde is one of the most common chemical procedures used to form the crosslinking of the collagen fibers. Nevertheless, the cytotoxicity associated with glutaraldehyde is a concern for clinical application [14,15,16,17]. Our study showed that BioMend Extend, the glutaraldehyde crosslinked membrane, had lower amounts of fibroblast growth on the membrane surface compared with the other membranes, and a statistically significant difference was seen in cell proliferation property between Jason and BioMend Extend membranes. Some previous studies have indicated more cytotoxicity from BioMend Extend when compared to Jason membrane [31], a result consistent with the findings of this study. These findings may warrant the use of native collagen membranes in cases with high concerns of wound dehiscence due to large grafts and high flap tension to allow better tissue integration during the first few days, which is usually the period of greatest risk due to the inflammatory phase of wound healing [30,34,35]. Nonetheless, many other studies have shown different cell types that are biocompatible with the chemically crosslinked collagen membranes such as BioMend^®^ extend, which may indicate that the tissue integration property of chemically crosslinked collagen membranes are clinically sufficient [19,36,37,38,39]. Furthermore, the origin of the collagen membranes did not seem to affect the ability of HGF to attach and proliferate on membrane surfaces, which is a contrasting finding compared to a study by Rothamel et al., where membranes of porcine origin showed greater tissue integration due to its lower immunogenicity and foreign body reaction compared to membranes of bovine origin [19].

The upper and lower surfaces of collagen membranes are the main focus of morphological study, since they impact many properties of the collagen membrane. Typically, collagen fibrils can form a crosslinked structure at a certain level for attachment and homing of the cells. Upper surfaces of all collagen membranes in SEM exhibited a densely packed collagen fibrils network creating a smooth surface with minimal porosities. Meanwhile, the lower surface of BioMend Extend membrane showed thick interwoven fibrils, which could be the result of glutaraldehyde treatment causing aggregation of the collagen fibrils [40]. Jason collagen membrane demonstrated clear fine fibril’s structure compared to the membranes of bovine origin, an observation that might be due to its pericardium origin as loosely packed collagen fibrils are usually observed in the pericardium, while the densely packed collagen fibrils commonly exhibited in the tendon tissue, which were observed in the GENOSS collagen membrane [41,42]. The nature of the fibrils used in different collagen membranes also affected their clinical handling properties, as porcine origin Jason membranes showed greater flexibility and were easy to attach to cell crowns™, while the bovine origin GENOSS and BioMend Extend membranes showed high stiffness and greater rebound and were prone to dislodge from cell crowns™ and tear when under higher application pressure [41,42].

This in vitro study has some limitations including the fact that only one kind of cell line was used. While this may represent general cell migration through different barrier membranes, it lacks other cell types such as mesenchymal stem cells, vascularized stem cells, and others. The presence of blood clots, inflammation, and possible infections can also be influential on the effectiveness of the membrane. Timing of cell adherence may also influence the results. The round morphology of the cells shown in Figure 1e,f depicts cells that are not in adhesion. One possible explanation is that the timing of penetration of the cells across the tested membrane may influence the cell morphology. Early penetration across the controls and Jason membranes provides the cells with time to attach on the membrane surfaces and hence, they present a spindle shaped morphology. Genoss and BioMend Extend membranes represent slower and later penetration, and hence not enough time for cell adhesion to be evident on the membrane surface. This finding is supported by various cell migration assay studies such as Lee, S.Y et al. [43]. Finally, this in vitro study only examined certain aspects of the cell migration associated with different barrier membranes. Other aspects such as cell differentiation and interaction with other cells are not examined in this study.

There are still many factors to consider when selecting collagen membranes for use in GBR. In fact, the barrier effects and tissue integration properties might be affected by many other issues. Our study proved that three commercially available collagen membranes can prevent gingival cell migration, and native collagen membranes allow better proliferation and growth of HGF compared to crosslinked collagen membranes. Additionally, the microstructure of the membranes is related to the clinical properties and handling characteristics of these membranes.

## 5. Conclusions

All collagen membranes appear to allow a small number of cells to migrate. The difference in barrier function ability in terms of prohibiting cell migration are similar with no different effect from cross-linked properties. Native collagen membranes allow a more even distribution of cellular proliferation compared to cross-linked collagen membranes. Collagen membranes of different origins and physical properties appear to be efficient in preventing cell migration and invasion. Native collagen membranes may provide a better surface for soft tissue integration in terms of gingival cell attachment and proliferation.

## Figures and Tables

**Figure 1 materials-16-01657-f001:**
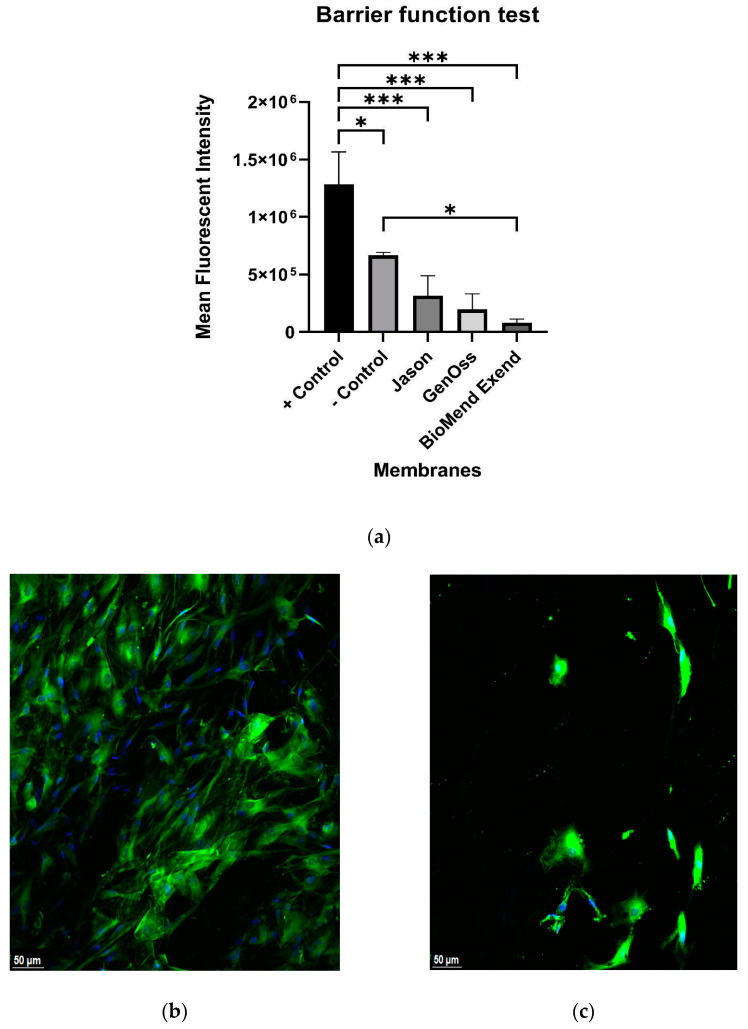
Barrier function test with 20X magnification under confocal microscopy. (**a**) All collagen membranes demonstrated the ability to impede cell migration, and there were no statistically significant differences between the membranes. Following the loading of HGF on membrane surfaces, the base of positive control PET membrane (**b**) displayed high quantities of penetrated HGF, whereas the base of negative control PET membrane (**c**), Jason membrane (**d**), GENOSS Collagen membrane (**e**) and BioMend Extend membrane (**f**) displayed very few penetrated HGF.

**Figure 2 materials-16-01657-f002:**
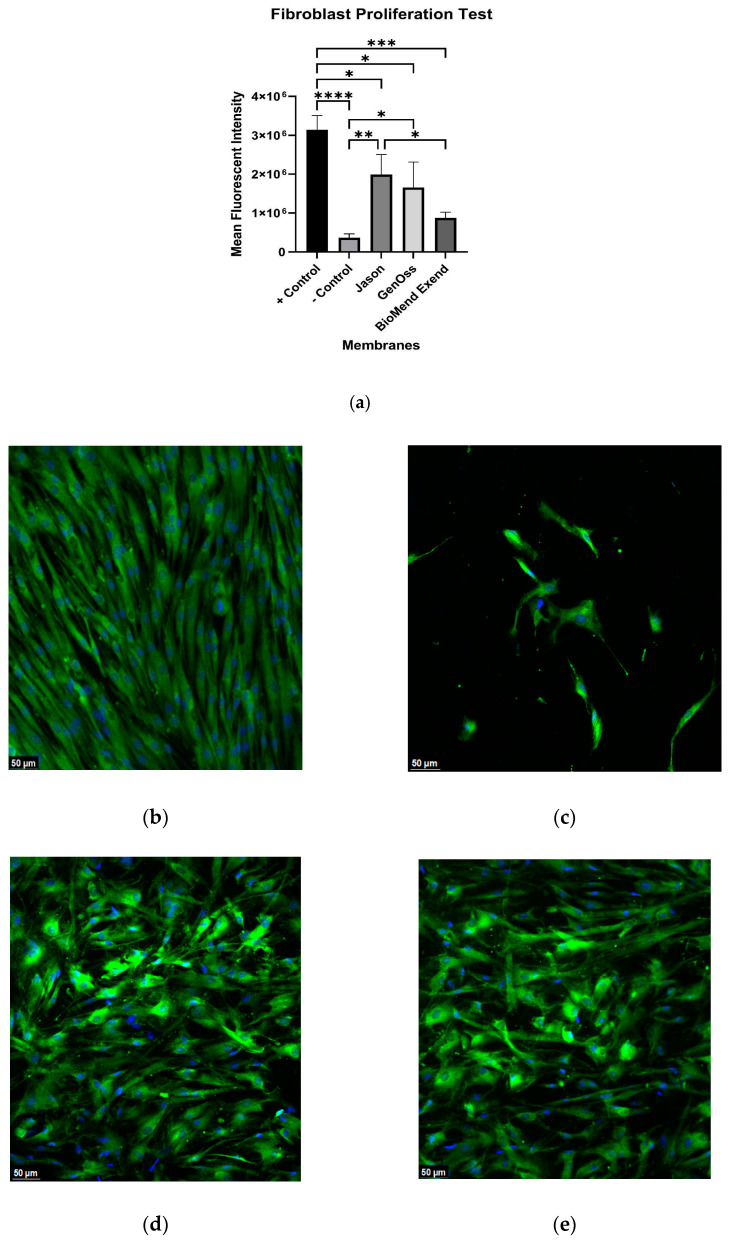
Fibroblast proliferation test with 20X magnification under confocal microscopy. (**a**) The fibroblast proliferation test revealed a significant difference between Jason and BioMend Extend membrane (*, *p* = 0.04), while there were no significant differences found between Jason and GENOSS membranes, and GENOSS and BioMend Extend membranes. Following the loading of HGF on membrane surfaces for 3 days, the positive control polyethylene terephthalate membranes (**b**) displayed an even dense proliferation of spindle shaped HGF across the entire membrane surface, while the negative control polyethylene terephthalate membranes (**c**) displayed poor HGF proliferation. Jason membrane (**d**) and GENOSS Collagen membrane (**e**) displayed dense and wide proliferation of HGF across the membrane surfaces, while Biomend Extend membrane (**f**) displayed uneven and scattered proliferation of HGF.

**Figure 3 materials-16-01657-f003:**
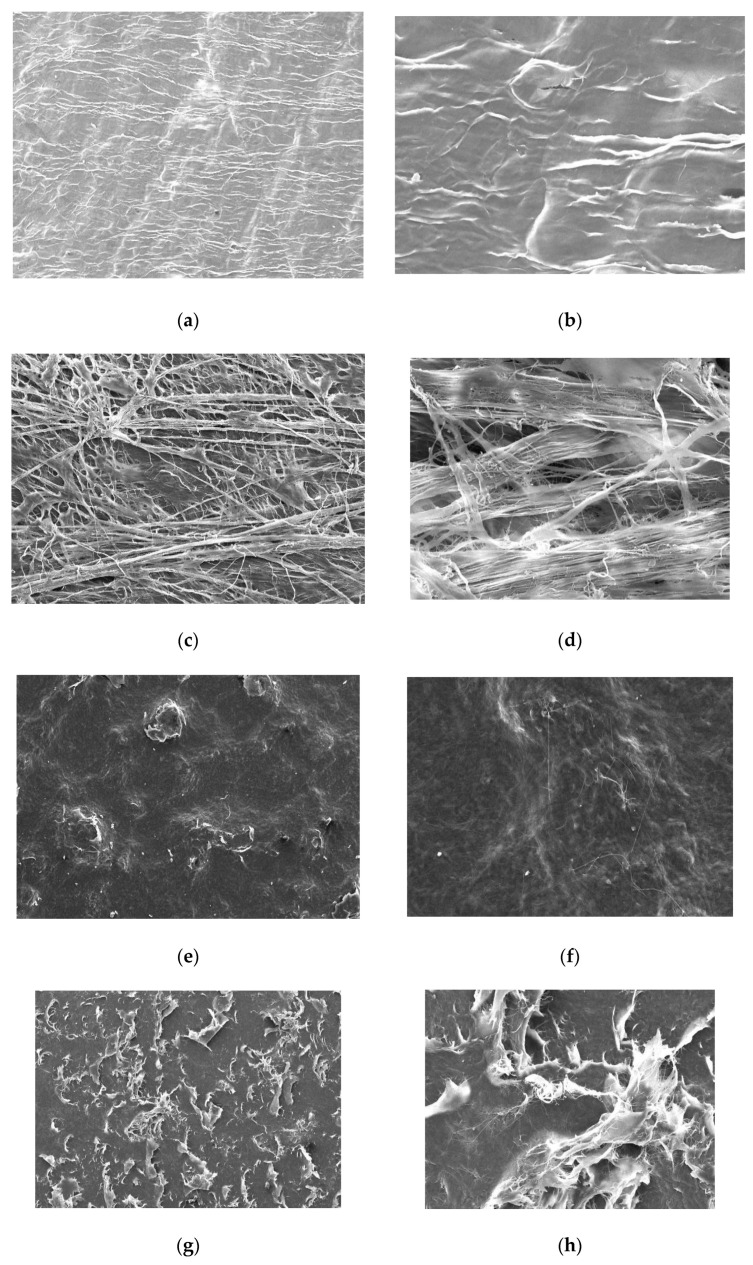
Scanning electron microscope photos: upper surface of Jason membrane at 100X (**a**) and 500X (**b**), base of Jason membrane at 100X (**c**) and 500X (**d**), upper surface of collagen membrane at 100X (**e**) and 500X (**f**), base of collagen membrane at 100X (**g**) and 500X (**h**), upper surface of BioMend Extend membrane at 100X (**i**) and 500X (**j**), base of BioMend Extend membrane at 100X (**k**) and 500X (**l**).

## Data Availability

Not applicable.

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
