# Peer review of "Occlusive and Proliferative Properties of Different Collagen Membranes—An In Vitro Study"

_materials, 2023, doi:10.3390/ma16041657_

Round 1

Reviewer 1 Report

Dear Authors,

Reported below are my comments and suggestions.

Best regards.

 In figures 1 and 2 please check that all scale bara are visible, and write in the captions the magnification used fo the acquisition. Further,  I suggest to the Authors to remove the tags with the name of the groups from each photo, moving and resizing, to give the reader the possibility to observe the overall images.

Observing figure 1, cell morphology in Jason membrane or control show differences compared with Genoss and BioMend where the cells are round in shape, typically of the cells that are not in adhesion or with a low viability. How do the Authors explain this point?

Moreover, I suggest the Author revise the discussion and conclusion completely because the project aims to evaluate the efficacy of several collagen membranes to block HGF cells, not epithelial cells. In the conclusion section, for example, the phrase “Collagen membranes of different origins and physical properties appear to be efficient in preventing epithelial invasion”, is not supported by the reported data. 

Author Response

REVIEWER #1:

In figures 1 and 2 please check that all scale bars are visible, and write in the captions the magnification used for the acquisition. Further, I suggest to the Authors to remove the tags with the name of the groups from each photo, moving and resizing, to give the reader the possibility to observe the overall images.

RESPONSE: Thank you for your recommendation. The scale bars are adjusted to be visible on all photos and the magnifications (20x) are added in the captions. The image tag labels were removed from the images as this information is already present in the caption under the figures.

TEXT CHANGE: “with 20x magnification under confocal microscopy” was added to the Figure legends.

Observing figure 1, cell morphology in Jason membrane or control show differences compared with Genoss and BioMend where the cells are round in shape, typically of the cells that are not in adhesion or with a low viability. How do the Authors explain this point?

RESPONSE: The authors agree that the round morphology of the cells in figures 1 e and f depict cells that are not in adhesion, and a possible explanation is the timing of penetration of the cells across the tested membranes. Early penetration across the controls and Jason membranes provides the cells with time to attach on the membrane surfaces and hence, present a spindle shaped morphology, while Genoss and BioMend Extend membranes represent slower and later penetration, and hence not enough time for cell adhesion on the membrane surface. This finding is supported by various studies regarding cell migration assays such as a study entitled “High Throughput 3D Cell Migration Assay Using Micropillar/Microwell Chips” by Lee, S.Y et al. in 2022. The authors have added this interesting point in the discussion under the barrier function test as well.

TEXT CHANGE: Additional clarification statement below is added in the Discussion.

“Timing of cell adherence may also influence the results. The round morphology of the cells shown in Figures 1 e and f depicting cells that are not in adhesion. One possible explanation is that the timing of penetration of the cells across the tested membrane may influence the cell morphology. Early penetration across the controls and Jason membranes provides the cells with time to attach on the membrane surfaces and hence, present a spindle shaped morphology. Genoss and BioMend Extend membranes represent slower and later penetration, and hence not enough time for cell adhesion to be evident on the membrane surface. This finding is supported by various cell migration assay studies such as Lee, S.Y et al. [41]”

Moreover, I suggest the Author revise the discussion and conclusion completely because the project aims to evaluate the efficacy of several collagen membranes to block HGF cells, not epithelial cells. In the conclusion section, for example, the phrase “Collagen membranes of different origins and physical properties appear to be efficient in preventing epithelial invasion”, is not supported by the reported data.

RESPONSE: We agree with the reviewers.

TEXT CHANGE: “epithelial” references were removed for most of the Discussion and Conclusions on cell migration. We only refer to as cell migration in a general term.

Reviewer 2 Report

The current study compares several collagen-based membranes for GBR at in vitro level. The manuscript is very well written and fulfills the purpose established by the researchers. However, given the extensive work carried out in the matter and the simple study design, the study has several points that must be addressed:

1) Introduction: The introduction is very well elaborated and includes the required information to the reader for the topic. However, there are two points that could be improved: 

a) The third paragraph reads: " A common [...] is collagen membrane that is made of natural polymers. [...]Natural collagen is extracted by decellularization and removal of antigenic components ...". This is confusing, since collagen-based products can be made of either purified collagen after extraction and scaffold fabrication or from decellularization and optional crosslinking (where the collagen is not extracted and also keep natural 3D microstructure). Authors refer to the later and should clarify this in the introduction.

b) There are several studies comparing membranes for GBR. Please recheck literature. Suggestions:

doi/10.1111/xen.12683 

doi: 10.3390/ma13030786 (already cited in the manuscript)

2) This reviewer has several concerns with the low proliferation observed in this study. Could these be due low viability (specially concerning Figures 1e and 1f? It is not logic that a decellularized native membrane prevents fibroblast proliferation (kills?), when using the same medium than a positive control of PET. A viability assay should be included. 

3) The barrier function test does not seem descriptive enough. This reviewer thinks that after seeding of fibroblast and an established time point, a transversal imaging (after histology) would be more descriptive.

4) Why proliferation is not measured on both surfaces of each material, when these are described in the SEM analysis?

Minor comment:

Figure 1 and 2 should be displayed similarly to Figure 3 for clearer observation for the reader and direct comparison. Also, the legends / labels of each sample should be smaller, they take too much field of view of the image.

Author Response

REVIEWER #2

The current study compares several collagen-based membranes for GBR at in vitro level. The manuscript is very well written and fulfills the purpose established by the researchers. However, given the extensive work carried out in the matter and the simple study design, the study has several points that must be addressed:

1) Introduction: The introduction is very well elaborated and includes the required information to the reader for the topic. However, there are two points that could be improved:

a) The third paragraph reads: " A common [...] is collagen membrane that is made of natural polymers. [...]Natural collagen is extracted by decellularization and removal of antigenic components ...". This is confusing, since collagen-based products can be made of either purified collagen after extraction and scaffold fabrication or from decellularization and optional crosslinking (where the collagen is not extracted and also keep natural 3D microstructure). Authors refer to the later and should clarify this in the introduction.

RESPONSE: We truly appreciate this insightful suggestion.

TEXT CHANGE: The following clarification statement was added in the Introduction.

“Collagen-based products can be made of either purified collagen after extraction and scaffold fabrication or from decellularization and optional crosslinking where the collagen is not extracted and also keep natural 3D microstructure. Different processing of collagen in theory may affect cellular and tissue healing responses.”

b) There are several studies comparing membranes for GBR. Please recheck literature. Suggestions:

doi/10.1111/xen.12683

doi: 10.3390/ma13030786 (already cited in the manuscript)

RESPONSE: Appreciate the suggestion.

TEXT CHANGE: The following reference is added into the introduction (new reference #12).

Capella-Monsonís H, Zeugolis DI. Decellularized xenografts in regenerative medicine: From processing to clinical application. Xenotransplantation. 2021 Jul;28(4):e12683. doi: 10.1111/xen.12683.

2) This reviewer has several concerns with the low proliferation observed in this study. Could these be due low viability (specially concerning Figures 1e and 1f? It is not logic that a decellularized native membrane prevents fibroblast proliferation (kills?), when using the same medium than a positive control of PET. A viability assay should be included.

RESPONSE: Thank you for your comments and raising a very interesting and valid point. The fibroblast proliferation tests are depicted in figure 2, while figure 1 displays the barrier function test. In terms of the viability of the cells, prior to loading the cells on each of the membranes, the cells were stained with CytoPainter Cell Proliferation Staining Reagent (Green Fluorescence), which only stains viable cells with intact cell membranes, and hence the non-viable cells are not stained and seen in the results (https://www.abcam.com/cell-proliferation-staining-reagent-green-fluorescence-cytopainter-ab176735.html). In addition, in this study, an equal number of cells were loaded in each membrane. Better proliferation could be observed in PET membranes due to 2 major differences between PET membranes and collagen membranes. Firstly, the 0.4 μm porosity on the surface of PET membranes provides conditions favorable for fibroblast adhesion, and subsequent proliferation as compared to the non-porous and compact collagen surfaces (also seen in the SEM results). Secondly, the PET membranes used in this study by Sigma-Aldrich are surface treated to enhance fibroblast adhesion and proliferation, as these membranes are used in studies to mimic in vivo cell adhesion and proliferation, a process the collagen membranes don’t undergo. The authors have included a link to the website stating the surface modifications on PET membranes to study cell adhesion and proliferation.

https://www.sigmaaldrich.com/TH/en/products/labware/cell-culture-and-cryogenics/millicell-cell-culture-inserts-and-plates

More importantly, the viable remaining cells may be a result lack of proliferation due to time require for cell attachment. Please see also the response to the Reviewer #1 comment.

TEXT CHANGE: A clarification statement is added (See also response to Reviewer #1)

“Timing of cell adherence may also influence the results. The round morphology of the cells shown in Figures 1 e and f depicting cells that are not in adhesion. One possible explanation is that the timing of penetration of the cells across the tested membrane may influence the cell morphology. Early penetration across the controls and Jason membranes provides the cells with time to attach on the membrane surfaces and hence, present a spindle shaped morphology. Genoss and BioMend Extend membranes represent slower and later penetration, and hence not enough time for cell adhesion to be evident on the membrane surface. This finding is supported by various cell migration assay studies such as Lee, S.Y et al. [41]”

3) The barrier function test does not seem descriptive enough. This reviewer thinks that after seeding of fibroblast and an established time point, a transversal imaging (after histology) would be more descriptive.

RESPONSE: The authors agree and have described the barrier function test in more details. After seeding of fibroblasts, the time point used in this study was set at 3 days due to two major reasons. 3 days post-operatively is critical as the wound inflammation peaks around this time, which makes the wound susceptible to dehiscence. Along with this, after multiple pilot experiments, the positive control PET membranes allowed for cell migration and penetration on the third day, and hence, the third day was chosen since cell migration could be seen at this time point in the transwell assay. A transversal imaging was not needed in this study due the thinness of the membranes, which did not allow for the collection of the fibroblasts in the middle layers. In addition, confocal microscopy was used to view the membranes layer by layer, and the fibroblasts were only detected on either surfaces of the membranes, and not in between the outer layers.

TEXT CHANGE: N/A

4) Why proliferation is not measured on both surfaces of each material, when these are described in the SEM analysis?

RESPONSE: Thank you for raising an interesting point. clinically, commercially available collagen membranes have a smooth outer surface, and a rough inner surface. This can also be seen in our SEM analysis. During GBR, the smooth outer surface is placed facing towards the soft tissues, while the rough inner surface is placed towards the bone. Hence, it is important for the smooth outer surface to enhance fibroblast proliferation to limit chances of potential wound dehiscence. For that reason, in our study, the fibroblast proliferation was viewed on the smooth outer surface of the collagen membranes, while the rough inner surface was viewed to detect penetrated fibroblasts.

TEXT CHANGE: N/A

Minor comment:

Figure 1 and 2 should be displayed similarly to Figure 3 for clearer observation for the reader and direct comparison. Also, the legends / labels of each sample should be smaller, they take too much field of view of the image.

RESPONSE: We have removed the labels completely to allow readers to clearly observe the images, and this information is in the caption under the figures. The authors agree that this gives the readers the chance to observe the images completely.

TEXT CHANGE: N/A

Round 2

Reviewer 1 Report

Dear Authors,

Thank you for the revisions, now the manuscript is improved.

Please replate figure 3 with a version without white boxes

Best  Regards. 

Author Response

REVIEWER #1

Thank you for the revisions, now the manuscript is improved.

Please replate figure 3 with a version without white boxes

RESPONSE: We appreciate the comment. The figure 3 is now without the white boxes.

Reviewer 2 Report

Authors have addressed many of the points raised by this reviewer. However, there are still some points that still must be addressed so the manuscript can be accepted.

11)      First, could authors submit the revised version without using the mark up tool from Word? It makes very confusing to follow up, especially the figures. Just a change in font color should be used to highlight updated sections of text.

22)      The answer to comment 3 should be included in the discussion. This clarification is important to the reader.

33)      Comment 4 answer. While the explanation to the use of only one side of the membrane is valid, authors should include it in the discussion / methods and provide some references. Most likely the effects observed are due to the presence of the basement membrane. This should be also discussed. Suggestions:

DOI: 10.1016/j.actbio.2013.09.006

DOI: 10.1089/ten.2006.12.519

44)      Please include a section in Methodology describing statistical analysis employed

Author Response

REVIEWER #2

Authors have addressed many of the points raised by this reviewer. However, there are still some points that still must be addressed so the manuscript can be accepted.

11)      First, could authors submit the revised version without using the mark up tool from Word? It makes very confusing to follow up, especially the figures. Just a change in font color should be used to highlight updated sections of text.

RESPONSE: All markups were removed. 

TEXT CHANGE: The additional texts are now highlighted in red color.

22)      The answer to comment 3 should be included in the discussion. This clarification is important to the reader.

RESPONSE: We appreciate the comment.

TEXT CHANGE: The following clarification was added in the Discussion.

“Note that in regards to the barrier membrane testing, after seeding of fibroblasts, the time point used in this study was set as a period of 3 days due to two major reasons. First, a three-day postoperative period is critical as the wound inflammation peaks around this time, which makes the wound susceptible to dehiscence. Second, after multiple pilot experiments, the positive control PET membranes allowed for cell migration and penetration on the third day, and hence, the third day was chosen since cell migration could be seen at this time point in the transwell assay. A transversal imaging was not needed in this study due the thinness of the membranes, which did not allow for the collection of the fibroblasts in the middle layers. Third, confocal microscopy used to view the membranes layer by layer demonstrated that the fibroblasts were only detected on either surfaces of the membranes, and not in between the outer layers.”

33)      Comment 4 answer. While the explanation to the use of only one side of the membrane is valid, authors should include it in the discussion / methods and provide some references. Most likely the effects observed are due to the presence of the basement membrane. This should be also discussed. Suggestions:

DOI: 10.1016/j.actbio.2013.09.006

DOI: 10.1089/ten.2006.12.519

RESPONSE: We appreciate the comment.

TEXT CHANGE: A discussion statement was added with the two new references

“The collagen membrane surfaces mimic epithelial basement membranes of extracellular matrix (ECM) as well as function as a scaffold that may play a role in facilitating fibroblastic attachment  [32,33]”

REFERENCE #32 Faulk DM, Carruthers CA, Warner HJ, Kramer CR, Reing JE, Zhang L, D'Amore A, Badylak SF. The effect of detergents on the basement membrane complex of a biologic scaffold material. Acta Biomater. 2014 Jan;10(1):183-93. doi: 10.1016/j.actbio.2013.09.006.

REFERENCE #33 Brown B, Lindberg K, Reing J, Stolz DB, Badylak SF. The basement membrane component of biologic scaffolds derived from extracellular matrix. Tissue Eng. 2006 Mar;12(3):519-26. doi: 10.1089/ten.2006.12.519. PMID: 16579685.

44)      Please include a section in Methodology describing statistical analysis employed

RESPONSE: We appreciate the comment.

TEXT CHANGE: The statistical section was added as follow:

“2.5 Statistical analysis

Intergroup comparison of barrier function test and fibroblast proliferation test of the three collagen membrane brands and control membranes were performed using one-way analysis of variance (one-way ANOVA) at a confidence interval of 95%, and post hoc analysis was done using Tukey’s multiple comparison test.”
